# Enhancing Shared Control for Telepresence in Dynamic Environment using Large Language Models

Manali Jain
*Dept. of Applied Mechanics and Biomedical Engineering*
*IIT Madras*
Chennai, India

Aditya Kanade
*Dept. of Electrical Engineering*
*IIT Madras*
Chennai, India

Bijo Sebastian
*Dept. of Engineering Design*
*IIT Madras*
Chennai, India

Manivannan Muniyandi
*Dept. of Applied Mechanics and Biomedical Engineering*
*IIT Madras*
Chennai, India

*Abstract*—Teleoperation in robotic systems encompasses three primary modes of control: full teleoperation, shared control, and autonomous operation. Full teleoperation allows human operators to have complete control over the robot, enabling real-time manipulation and decision making. Shared control, a hybrid approach, integrates elements of both teleoperation and autonomous control, permitting human intervention in specific scenarios while maintaining a degree of autonomous functionality. Autonomous operation relies entirely on the robot's decision-making algorithms to perform tasks without human input. Although shared control has proven effective in static environments, recent studies indicate that its benefits diminish in dynamic settings due to the increased cognitive load on the human operator and the frequent need to switch between modes. The advent of multimodal large language models (LLMs) such as GPT-4 and Gemini has significantly advanced visual scene understanding and language-based reasoning. These capabilities can enhance shared control systems by allowing operators to act as global planners and provide natural language commands, reducing the need for constant switching. This paper proposes a novel approach that combines language-driven machine learning models with shared control frameworks to improve human-robot interaction in both static and dynamic environments. We develop a language-model-guided shared control mechanism and evaluate its performance across various settings. Results from both qualitative feedback and quantitative metrics demonstrate that our LLM-based shared controller successfully reduces operator cognitive burden while improving overall task performance.

*Index Terms*—shared control, LLM, teleoperation, user perception

## I. INTRODUCTION

Teleoperation in robotic systems involves three main modes of control: full teleoperation, shared control, and autonomous operation [1]. Full teleoperation grants the human operator complete control over the robot, providing real-time manipulation and decision-making capabilities. Shared control, a hybrid approach, combines elements of both teleoperation and autonomous control, allowing human intervention in specific scenarios while the robot maintains a degree of autonomous functionality [2]–[9]. Autonomous operation, on the other hand, relies entirely on the robot's decision-making algorithms to navigate and perform tasks without human input.

Shared control has been shown to be effective in static environments [10], where the stability and predictability of the environment allow efficient human-robot collaboration. However, a recent study indicates that the benefits of shared control diminish in dynamic environments, where obstacles and environmental conditions may change unpredictably [11]. This study did not find a clear preference for a mode of control in such settings, suggesting that the increased cognitive load and the frequent need to switch between autonomous and teleoperated modes may reduce the overall effectiveness of shared control in dynamic scenarios.

Shared control in robotic systems uses human guidance for complex maneuvering scenarios, positioning humans as global planners. However, in dynamic environments where obstacles can appear unexpectedly, this approach can increase cognitive load and frustration for the human operator compared to direct teleoperation. Human operators, hereafter referred to as controllers, might find it easier to handle constant stimuli in a dynamic setting with a fully teleoperated robot, rather than frequently switching between autonomous and teleoperated modes.

The advent of multimodal large language models (LLMs) such as GPT-4 [12] and Gemini [13] has significantly advanced visual scene understanding and language-based reasoning. These capabilities can be harnessed to improve shared control systems by allowing operators to provide natural language commands, which could reduce the need for constant switching. Acting as substitutes for traditional shared controllers, multimodal LLMs can interpret and execute natural language instructions in real time. This approach not only mitigates the cognitive load on the human operator, but also enhances the overall efficiency and safety of robotic operations in dynamic environments.

In this paper, we evaluate the above proposed approach to combine language-driven machine learning models with shared control frameworks, towards improving human-robot interaction in both static and dynamic settings. To this extent, we present a shared control mechanism guided by the language model and assess its performance in different environments.

Our study aims to demonstrate the effectiveness of this model in alleviating operator stress and to examine qualitative differences in task execution between static and dynamic settings.

## II. RELATED WORK

### A. Shared Control Mechanism in Teleoperation

In robotic telepresence, a user remotely operates a robot situated in a different environment. The most common setup in research involves a mobile robot equipped with a camera, screen, and control system [14]. Shared control, widely utilized in various robotic applications [15]–[18], involves collaboration between a human and a robot to achieve specific goals. It incorporates techniques such as policy mixing, intent prediction, and dynamic adaptation of autonomy levels to improve robot-user collaboration [19]. This approach blends manual and autonomous modes, utilizing safeguards such as collision detection and context-aware arbitration to improve navigation performance and reduce operator workload [20], [21]. Additionally, shared control methods combine obstacle handling, akin to safeguard approaches, with autonomous planning to minimize the need for active user intervention [22]. While studies have validated the efficacy of shared control in teleoperation for static virtual reality environments, its performance in dynamic scenarios remains underexplored, requiring further investigation [23]. In this context, we present a shared control mechanism guided by a language model, designed to alleviate operator stress and enhance task execution. Our study evaluates its effectiveness and explores qualitative differences in performance across static and dynamic environments.

### B. LLMs for Robotics

Large language models (LLMs) have emerged as powerful tools in natural language processing, demonstrating remarkable capabilities in few-shot and zero-shot learning, commonsense reasoning, and multistep computations [24]–[29]. These strengths have inspired recent efforts to integrate LLMs into robotics, unlocking novel applications and improving task execution. LLMs have been leveraged for various robotic applications, such as using pretrained skills for context-aware actions [30], enabling long-horizon reasoning for sequential tasks [31], applying semantic translation for actionable plans [32], and incorporating error feedback to enhance executability [33]. Further advancements include language-based navigation without fine-tuning [34], few-shot prompting for solving planning problems using pretrained large language models [35], integrating commonsense knowledge for 3D scene understanding [36], generating robot policy code from natural language commands [37], and language-conditioned meta-learning for adaptive tool manipulation [38] and leveraging multi-modal instructions for robotic manipulation through Python-based perception, planning, and action loops [39], and integrating vision-language models into end-to-end robotic control for improved generalization and emergent semantic reasoning [40]. While these studies highlight the use of LLMs to improve the intelligence of robotic systems, our approach extends this

by showcasing how LLMs can facilitate shared control in dynamic environments.

### C. Foundation Models in Robotics

Foundation models have increasingly been explored for robotic applications, leveraging their advanced capabilities in reasoning, planning, and multi-modal integration to enable more intelligent and adaptable robotic systems. Recent approaches include manipulator trajectory synthesis using affordance reasoning [39] and mapping multi-modal instructions into sequential actions for robotic manipulation tasks, as demonstrated in Instruct2Act [41]. Notable progress has been achieved with frameworks like VIMA-BENCH, where a multi-modal prompt-based learning model set a new state-of-the-art [42]. For more complex tasks, Jin et al. proposed a closed-loop multi-modal planning model, improving success rates in multi-step reasoning for robotic manipulation [43]. Myers et al. enhanced policy learning by integrating language with image-based goal representations, enabling efficient alignment of instructions with visual objectives [44]. Similarly, Driess et al. introduced PaLM-E, a multi-modal language model that integrates sensor data and text for grounded reasoning, excelling in tasks like robotic manipulation, visual question answering, and captioning [45]. Other innovations include repurposing code-writing LLMs to autonomously generate robot policy code for reactive and waypoint-based policies [46], and leveraging the semantic knowledge of language models through guided decoding strategies to solve complex, long-horizon embodiment tasks by aligning language and grounded models [47]. These developments collectively highlight the transformative role of foundation models in enhancing the intelligence of robotic systems, advancing shared control, planning, and task execution in different environments.

## III. HYPOTHESES

The integration of advanced language models into shared control frameworks represents a significant advancement in enhancing human-robot interaction. These systems facilitate seamless communication between operators and robots while offering the potential to reduce cognitive strain in complex environments. To evaluate shared control systems, it is essential to consider both user preferences and perceptions of workload, as these factors play a crucial role in determining their adoption and effectiveness. By analyzing user satisfaction and task performance across different control paradigms, we can identify methods that achieve an optimal balance between human input and autonomous functionality.

**H1**: Participants prefer the LLM-Based Shared Control (LSC) over Regular-Based Shared Control (RSC), as indicated by their responses to direct preference questions.

**H2**: The perceived workload is lower under LSC compared to RSC. This is assessed by directly asking participants which condition they found easier and using the NASA-TLX questionnaire after each condition.

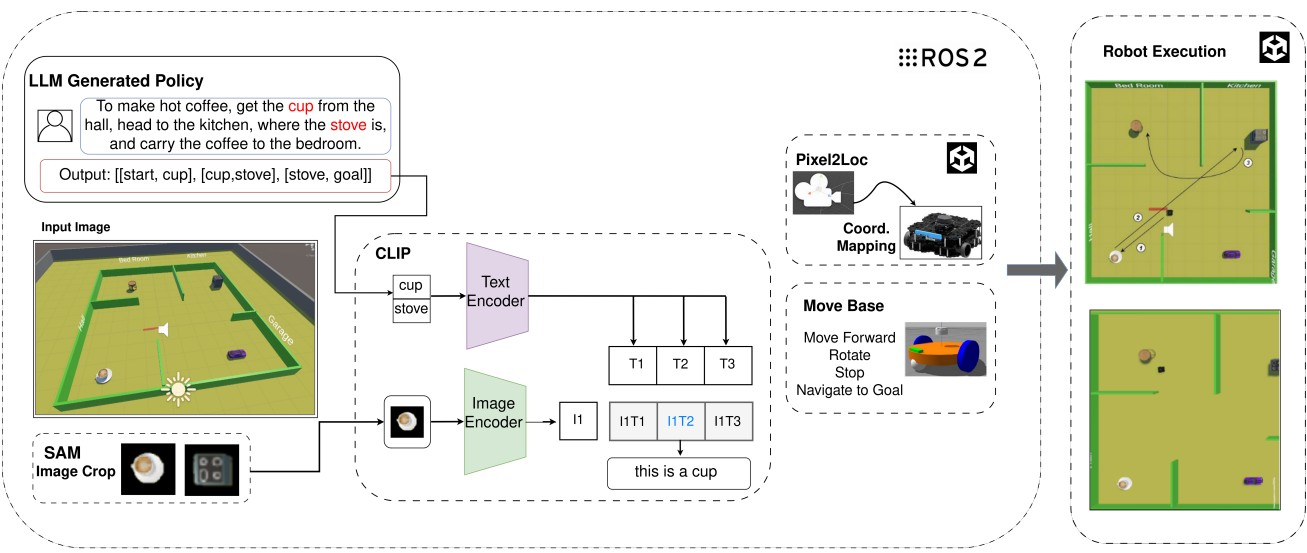

Fig. 1. The paradigm of our proposed framework begins with the user providing task instructions, which are processed by a large language model (LLM). The LLM interprets the command and converts it into a structured sequence of discrete actions. These actions are further refined through the use of visual foundation models, which analyze the environment by recognizing object semantics. Leveraging this environmental understanding, the framework generates a set of feasible actions tailored to the task. These actions are then transmitted to Unity, where pixel-to-location mapping enables the robot to execute the task.

## IV. FRAMEWORK

The hypotheses were tested in a Unity-based simulated environment. The telepresence robot, equipped with a mobile base, used ROS and the Nav2 [48] framework for autonomous navigation, integrated with controllers and guided by LLM-based instructions, building on the setup from [11]. This research enhances the framework by incorporating advanced language models into shared control systems, aiming to improve human-robot interaction and task efficiency. The enhanced framework is systematically designed, with each component detailed in the following sections.

***Overview of the System*** - Consider a scenario where a user issues a complex command:

> *"To make hot coffee, get the cup from the hall, head to the kitchen, where the stove is, and carry the coffee to the bedroom."*

Translating such a command into robot actions involves three key components:

1. Natural Language Understanding: Parsing and interpreting the intent of the user. 2.Action Sequence Generation: Converting the interpreted command into a structured sequence of discrete actions. 3.Robot execution: Performing the generated action sequence in the physical environment [41], [49].

### A. Unified Architecture for Natural Language to Robot Action

We now outline our pipeline that transforms high-level language commands into executable tasks within a simulated environment. The pipeline integrates four major components, as shown in Fig1.

- **Large Language Model (LLM)**: Translates natural language into an ordered list of actions.

- **Segment Anything Model (SAM)** [50]: Segments objects in the robot's panoramic camera view.
- **CLIP** [51]: Associates these segmented objects with meaningful labels.
- **pixel2loc**: Maps 2D pixel coordinates to 3D positions in the Unity environment.

We use the Gemini[1] LLM model [13] to parse natural language instructions. In order to align the LLM response with our objective, we also add a few in-context examples. Following is an outline of the examples we used for our prompts:

> *Input: 'Pick up the book from the desk and place it on the shelf.'*
> ```
> Output: [[start, book], [book,
> shelf], [shelf, goal]]
> ```
>
> *Input: 'Water the plant in the living room, then return the watering can to the kitchen.'*
> ```
> Output: [[start, plant], [plant,
> goal]]
> ```

Given these examples, the model can then process new commands:

> *Input: 'To make hot coffee, get the cup from the hall, head to the kitchen, where the stove is, and carry the coffee to the bedroom.'*
> ```
> Output: [[start, cup], [cup,
> stove], [stove, goal]]
> ```

Actions generated by the LLM reference specific objects or locations. To identify these objects in the virtual camera

---

[1]Experiments were conducted using Google's Gemini 1.5 Flash model accessed through the Google Cloud AI Platform.

feed, the Segment Anything Model (SAM) extracts precise segmentation masks for each object. Subsequently, CLIP [51] links the segmented regions to labels provided by the LLM, ensuring accurate object identification and reducing ambiguity in cluttered or dynamic environments.

Finally, the `pixel2loc` function employs depth information to convert the 2D pixel coordinates of segmented objects into 3D locations within the Unity environment. By aligning the segmented image data with real-world (simulated) geometry, the robot can navigate and interact with the correctly identified objects.

Analytically, our LLM-based navigation framework is defined as follows: The robot starts at an initial position $S$ and captures an image $I \in \mathbb{R}^{H \times W \times 3}$ where $H$ is the height and $W$ is the width ($640 \times 480$), along with a depth map $D \in \mathbb{R}^{H \times W}$.

A natural language command $H$ is provided by a human, describing the global navigation task.

The global plan described in $H$ is decomposed using a Large Language Model (LLM) into a sequence of keypoints:

$$\mathcal{P} = \{[S, K_1], [K_1, K_2], \ldots, [K_n, G]\}, \quad \mathcal{K} = \{K_1, K_2, \ldots, K_n\}$$

where $G$ is the goal.

The input image $I$ is segmented into $k$ constituent objects $\mathcal{O} = \{O_1, O_2, \ldots, O_k\}$ using the Segment Anything Model (SAM). Each object $O_i$ is represented as:

$$O_i = \text{Region}(I, R_i), \quad R_i \subseteq [1, H] \times [1, W]$$

where $R_i$ is the pixel region corresponding to the $i$th object.

Each object $O_i$ and keypoint $K_j$ are encoded using CLIP to obtain feature vectors:

$$f(O_i) = \text{CLIP}_{\text{image}}(O_i), \quad f(K_j) = \text{CLIP}_{\text{text}}(K_j)$$

The most similar object for a given keypoint $K_j$ is identified using cosine similarity:

$$i^* = \arg\max_i \cos\big(f(O_i), f(K_j)\big)$$

The centroid pixel position of the matched object $O_{i^*}$ in the image $I$ is:

$$c_{i^*} = \text{Centroid}(R_{i^*}) = (x_{i^*}, y_{i^*})$$

Using the depth map $D$, the centroid pixel position is reprojected to 3D coordinates:

$$C_{i^*} = \text{pixel2loc}(c_{i^*}, D) = (X_{i^*}, Y_{i^*}, Z_{i^*})$$

This process is applied for all keypoints $K_j \in \mathcal{K}$, resulting in a sequence of 3D coordinates:

$$\mathcal{P}_{\text{3D}} = \{[S, C_{K_1}], [C_{K_1}, C_{K_2}], \ldots, [C_{K_n}, G]\}$$

The robot navigates sequentially through these 3D keypoints, executing the planned trajectory. Thus, by combining language understanding (LLM), visual segmentation (SAM), semantic labeling (CLIP), and 3D mapping (`pixel2loc`), this architecture seamlessly translates human instructions into robotic actions. The system is robust to variations in household

---

**Algorithm 1** Pixel to World Location (Pixel2Loc)
**Require:**
1: Camera Intrinsics $\mathbf{K} \in \mathbb{R}^{3 \times 3}$
2: Camera-to-World Transformation $\mathbf{T}_{\mathcal{WC}} \in SE(3)$
3: Pixel Coordinates $\mathbf{u} = [u, v]^\top$
4: Depth at Pixel $d = \mathbf{D}[v, u]$
**Ensure:**
5: 3D World Coordinates $\mathbf{p}^{\mathcal{W}} \in \mathbb{R}^3$
6: **function** PIXEL2LOC($\mathbf{K}, \mathbf{T}_{\mathcal{WC}}, \mathbf{u}, d$)
7:    **(1) Form homogeneous pixel coordinates:**
8:    $\tilde{\mathbf{u}} \leftarrow [u, v, 1]^\top$
9:    **(2) Back-project to camera coordinates:**
10:    $\mathbf{p}^{\mathcal{C}} \leftarrow d\,\mathbf{K}^{-1}\tilde{\mathbf{u}}$
11:    **(3) Transform to world coordinates:**
12:    $\mathbf{p}^{\mathcal{W}} \leftarrow \mathbf{T}_{\mathcal{WC}} \begin{bmatrix} \mathbf{p}^{\mathcal{C}} \\ 1 \end{bmatrix}$
13:    **return** $\mathbf{p}_{1:3}^{\mathcal{W}}$   *(the first three components)*
14: **end function**

---

layouts and object types, while real-time performance supports dynamic scene changes. Challenges remain in handling ambiguous language [52] and cultural or household-specific conventions [53], yet the presented pipeline substantially lowers the cognitive load on human operators and provides a foundation for more advanced human-robot interaction.

### B. Task Execution in Unity Environment

The 3D positions and associated commands are transferred to the Unity environment, enabling the robot to navigate and execute the tasks specified in the natural language instructions. The Unity environment dynamically updates to reflect the robot's movements and interactions, providing real-time feedback to the user.

This feedback loop is crucial to ensure that robot actions align with user expectations and to make any necessary adjustments. By integrating visual and language models with the Unity environment, the system offers a seamless and intuitive user experience, enabling the efficient execution of complex tasks through natural language commands.

## V. METHODOLOGY

### A. Procedure

Participants engaged in two experimental conditions, corresponding to both $LSC$ and $RSC$. Upon arrival, they were greeted by a researcher and asked to sign a consent form. They were pre-screened for nausea or headaches before starting the experiment. The experimenter demonstrated how to wear the head-mounted display (HMD) and provided general instructions.

As shown in Fig.2, the task instructions for the static environment were as follows: *"Prepare hot coffee: collect the cup from the hall, go to the kitchen near the stove, and bring the coffee to the bedroom."*

For the dynamic environment, as shown in Fig.3, the instructions for $LSC$ were: *"There is an emergency, and the doctor*

*requires an important book in room ER-205. First, collect the book from the reception area, then go to the first aid kit in the staff room, and deliver it to ER-205."* The task in the dynamic environment in ER-206 becomes more complex and dynamic due to the movement of people in the room.

Participants briefly practiced the controls, completed the tasks, and then filled out a questionnaire about their experience.

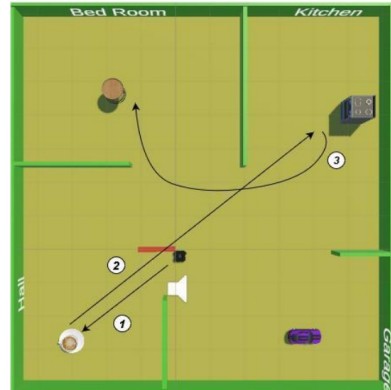

Fig. 2. Task in static environment.

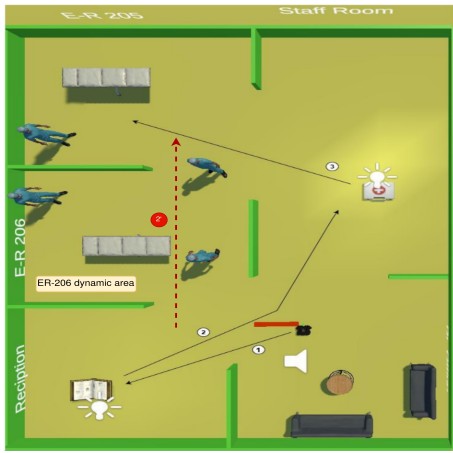

Fig. 3. Task in the dynamic environment.

## B. Measures and Participants

This user study involved 15 participants (8 men, 7 women) from various departments of a technical university. Of these, 31.8% had used VR frequently, 31.8% had used it a few times, 18.2% once or twice a year, and 18.2% had never used VR.

Participants performed tasks in both static and dynamic environments using two control methods: LSC and RSC. Objective measures such as task completion time and distance traveled were recorded, while subjective assessments were gathered via the NASA-TLX questionnaire to evaluate perceived effort and frustration. Participants also provided feedback on their preferences and ease of use for each method through open-ended questions, which explored the reasons behind their choices. These measures aimed to comprehensively assess the usability and effectiveness of LLM-powered shared control versus traditional methods in varying task environments.

## VI. RESULTS

### A. Confirmatory analysis

The Fig.4 the distributions of participants responses to the forced-choice questions regarding preference and ease of use. When asked "Which control method did you prefer?" 11 out of 15 participants (74%) selected the LSC condition in a static environment, while in a dynamic environment, 15 out of 15 participants (100%) selected the LSC condition. An exact binomial test with Clopper-Pearson 95% confidence intervals was performed, indicating that the LSC condition was significantly preferred compared to the RSC condition in the dynamic environment ($p < 0.01$), while in the static environment, the preference was not statistically significant ($p = 0.302$). This suggests that in the dynamic environment, participants' preference for the LSC condition was strong and unlikely to be due to random chance, whereas in the static environment, there was no strong evidence of a significant preference for the LSC condition over the RSC condition.

Similarly, when asked "Which control method felt easier?" 15 out of 15 participants (100%) selected the LSC condition in both static and dynamic environments, as shown in Fig.4. An exact binomial test with Clopper-Pearson 95% confidence intervals was performed, indicating that the LSC condition was significantly preferred compared to the RSC condition in the static and dynamic environment ($p < 0.01$) over RSC.

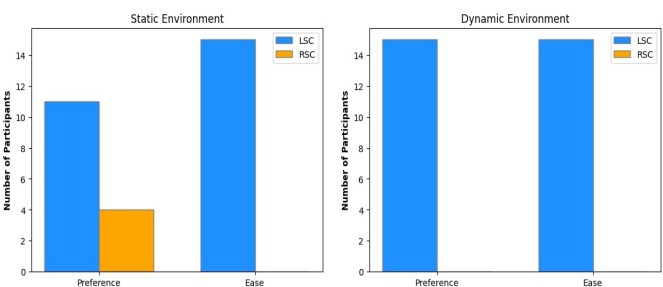

Fig. 4. Comparison of user preference and ease of use between static and dynamic environments for RSC and LSC Control Modes

We compared the differences in NASA-TLX workload scores, as shown in Fig.5, and performed a Wilcoxon Signed-Ranks test to analyze the TLX scores for effort in static environments: $RSC$ (Mdn = 33.33) and $LSC$ (Mdn = 5). The test indicated that $LSC$ elicited statistically significantly lower effort scores compared to $RSC$ ($Z = -3.455, p < 0.01, r = -0.774$).

Similarly, a Wilcoxon Signed-Ranks test was conducted to compare the TLX scores for frustration in static environments: $RSC$ (Mdn = 32.66) and $LSC$ (Mdn = 19.33). The test revealed that $LSC$ also elicited statistically significantly lower frustration scores compared to $RSC$ ($Z = -3.399, p < 0.01, r = -0.761$).

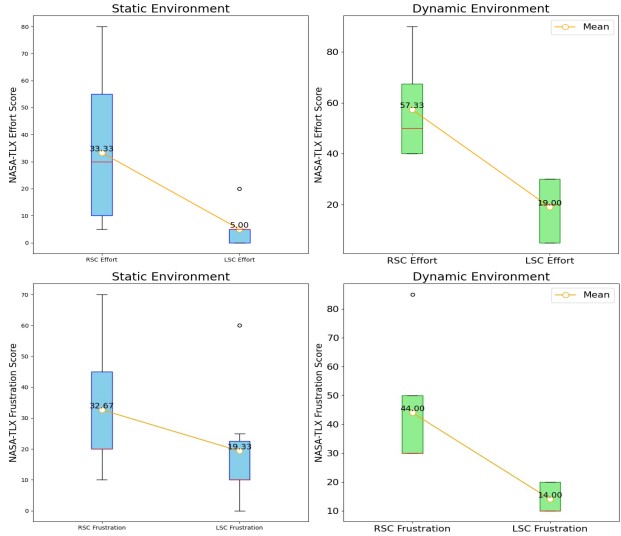

Fig. 5. Comparison of NASA TLX scores among static and dynamic environments based on different control modes RSC and LSC - (a) Effort (b) Frustration.

These findings suggest that the LSC is less demanding in terms of both frustration and effort, which has important implications for designing environments that reduce workload and enhance performance in static environments.

### B. Exploratory analysis

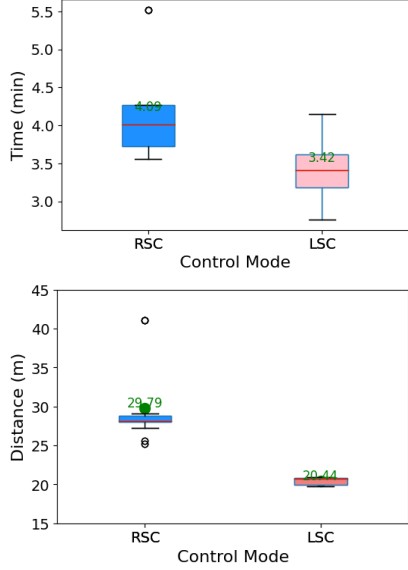

Fig. 6. Quantitative measures - Static Environments - (a) Time to completion of tasks (blue), (b) Total Distance traveled (pink)

*1) Static Environment:* The task completion time data was analyzed to determine if one method resulted in faster task completion. A one-way ANOVA was performed on the task completion time (in minutes), as shown in Fig.6, and revealed a statistically significant result: $F(3.343, 9.597) = 9.752$, $p = 0.04$. We observed that $RSC$ ($M = 4.088$) performed

significantly worse, while $LSC$ ($M = 3.421$) performed considerably better.

Similarly, an ANOVA was conducted to analyze the impact of control mode on the overall distance (in meters) covered by the robot to reach multiple targets, as shown in Fig.2. The analysis revealed a statistically significant result: $F(656.857, 312.485) = 58.857$, $p < 0.001$. We observed that $RSC$ ($M = 29.79$) performed significantly worse, while $LSC$ ($M = 20.43$) performed considerably better.

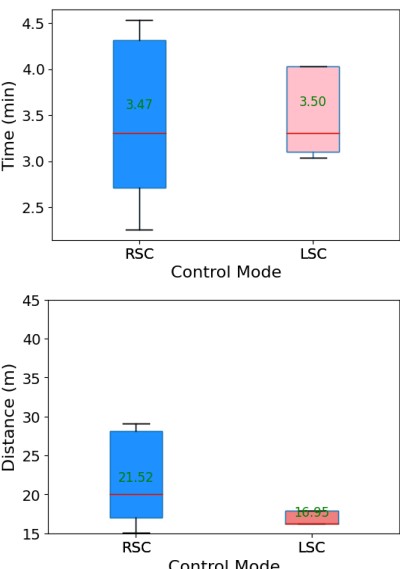

Fig. 7. Quantitative measures - Dynamic Environments - (a) Time to completion of tasks (blue), (b) Total Distance traveled (pink)

*2) Dynamic Environment:* A one-way ANOVA was conducted to analyze task completion time (in minutes), as shown in Fig.7, and revealed no statistically significant difference: $F(0.004, 17.811) = 0.007$, $p = 0.935$. We observed that $RSC$ ($M = 3.4671$) performed similarly to $LSC$ ($M = 3.490$). Given the large p-value, the task completion times for both methods are comparable, with any observed differences likely due to random variation. Thus, both control methods performed equivalently in terms of task completion time.

In contrast, the impact of the control modes on the total distance traveled by the robot to reach multiple targets, as depicted in Fig.7, was found to be statistically significant: $F(158.79, 508.68) = 8.470$, $p = 0.006$. The analysis revealed that $RSC$ resulted in a significantly greater distance traveled ($M = 21.535$) compared to $LSC$ ($M = 16.934$).

*3) Qualitative:* The open-ended data was analyzed using the thematic analysis method with an inductive approach. Responses to the open-ended question about why participants preferred each control method were examined. The most frequently cited reason for preferring LSC in both environments was that it was *"less demanding."* For participants who preferred RSC in the static environment, the primary factor was *"having more control."* However, in the dynamic environment, *"having more control"* was associated with increased workload and cognitive load.

In response to the open-ended questions about why a particular control method felt easiest to use, the majority of comments for LSC in static environments highlighted that it was *"less demanding,"* followed by reasons such as *"no switching"* and *"less collision."*

In dynamic environments, most users found the method easier to use due to the "no collision" feature in areas like the first aid kit, as participants tended to avoid crowded areas such as ER-206 (fig. 3). This was followed by reasons such as *"less demanding"* and *"no switching."*

## VII. DISCUSSION

This study compared the efficacy of LSC and RSC methods in both static and dynamic environments, revealing significant insights into the future of human-robot interaction interfaces. The results demonstrated distinct advantages of LSC, particularly in dynamic environments, though with nuanced trade-offs across different scenarios.

A fundamental trade-off emerged between perceived control and operational efficiency. While RSC provided users with a greater sense of direct control in static environments, this came at the cost of increased cognitive and workload demands, particularly in dynamic scenarios. LSC, conversely, achieved a more balanced optimization of control and efficiency through its unique human-as-global-planner architecture. In this system, users provide natural language commands describing their desired trajectory plan. These commands are then decomposed into discrete action states by an LLM, while CLIP and SAM models identify relevant navigational keypoints from the scene image. While local path planning between keypoints utilizes the traditional A* algorithm, the human effectively shapes the global navigation strategy from source to goal. This hierarchical approach enabled more efficient navigation around dynamic obstacles, reducing the need for frequent trajectory re-planning.

The performance metrics strongly supported LSC's advantages. In static environments, LSC demonstrated superior performance with 16.37% faster completion times and 31.04% shorter distances traveled compared to RSC. In dynamic environments, while task completion times were comparable between methods, LSC achieved 21.28% shorter travel distances, indicating more efficient path planning and obstacle avoidance. These improvements were further supported by user experience feedback.

The qualitative feedback consistently highlighted LSC's intuitive design, with 100% of participants preferring LSC in dynamic environments. Although 67% favored LSC in static settings ($p > 0.05$), this preference was not statistically significant.

NASA-TLX scores provided further evidence of LSC's benefits, particularly in reducing cognitive burden. In static environments, LSC showed a 40.79% reduction in frustration levels and a 85% decrease in effort compared to RSC ($p < 0.01$). This reduced cognitive load is particularly valuable for prolonged or stressful operations, where minimizing operator fatigue is crucial for maintaining performance. Aligning with our initial hypothesis, these results suggest that natural language-based control interfaces can effectively bridge the gap between human strategic thinking and robotic execution, particularly in complex, dynamic environments.

These findings have important implications for the design of robotcontrol and human-robot interaction systems. While LSC demonstrates clear advantages in dynamic or complex environments, the enhanced sense of control offered by RSC in static settings suggests potential value in hybrid approaches. Future research should explore the development of adaptive systems that can seamlessly transition between LSC and RSC based on environmental complexity, conduct long-term studies examining operator fatigue and performance adaptation over extended use periods, integrate more sophisticated natural language understanding to handle increasingly complex navigation scenarios, and apply this approach to diverse robotic platforms and task domains.

In conclusion, this study provides compelling evidence for the advantages of LLM-based shared control, particularly in dynamic environments. By incorporating intuitive natural language control schemes, robotic systems can achieve improved human-robot interaction while reducing cognitive load and enhancing operational efficiency. These insights contribute to the ongoing evolution of robotic control interfaces, with potential applications spanning from search-and-rescue missions to space exploration.

## VIII. CONCLUSION

In this paper, we introduced LLM-based Shared Control (LSC), a novel approach for telepresence robots that incorporates user input through text commands to influence robot trajectories. Our user study, conducted in virtual reality, across both static and dynamic environments, required participants to navigate a robot to multiple targets. The results showed that participants generally preferred LSC for its ease of use, particularly in dynamic environments, where it reduced frustration and effort compared to Regular Shared Control (RSC). While some users favored switching control methods in static environments, citing a greater sense of control, no statistically significant difference was found in the perceived feeling of control between methods. Notably, in dynamic environments, LSC led to decreased teleportation usage, indicating improved navigation efficiency. These findings highlight the potential of LSC as an effective control method for telepresence robots, especially in complex, dynamic settings where it reduces cognitive load and enhances the user experience.

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
