# OpenReview forum: "Enhancing Shared Control for Telepresence in Dynamic Environment using Large Language Models"
_humanrobotinteraction.org/HRI/2025/Workshop/VAM — HRI 2025 Workshop VAM Submission_

### Official Review · Reviewer_zWmU · 2025-02-28

**Rating:** 6
**Confidence:** 5

**Review:**

VAM-HRI Workshop Paper Review
Paper Information
Title: Enhancing Shared Control for Telepresence in Dynamic Environment using Large Language Models (LLMs)
Page Length: 7 pages plus two pages of references

Overall Recommendation
Weak reject - borderline rejection due to limited VR focus.

Description Summary
This paper presents an integration of LLMs with shared control for telepresence robots. While technically sound and innovative in its LLM integration, the paper uses VR primarily as an evaluation platform rather than making substantial contributions to virtual/augmented reality research. The work demonstrates reduced cognitive load and improved task performance, but how this work fits within major VAM-HRI themes is tangential.

Relevance to VAM-HRI Themes
Primary Theme Alignment
Central theme: Immersive systems for remote teleoperation,
Strength of theme alignment: Moderate to Weak,
Justification: While the paper uses VR for telepresence control, it focuses primarily on LLM integration and natural language processing. The virtual reality aspect is presented within the paper as a convenient choice rather than a core research contribution.

Secondary Themes
•        AR/VR for robot learning,
•        Natural language grounding,
•        Architectures for AR/VR-based HRI.

The theme integration emphasises language models and control systems over virtual/augmented reality aspects.

Technical Merit
Strengths
•        Novel integration of LLMs, CLIP, and SAM,
•        Statistical validation of results,
•        Clear experimental methodology,
•        Well-designed comparison between control methods.

Areas for Improvement
•        Limited VR hardware specifications and implementation details
•        The sample size (15 participants) needs justification,
•        Missing discussion of VR’s impact on control performance
•        Unclear environmental complexity considerations
•        No exploration of VR-specific challenges or benefits

Methodology Assessment
•        Research approach: Well-structured, but VR integration seems incidental,
•        Technical soundness: Strong LLM implementation, weak VR consideration,
•        Validation approach: Good general metrics, lacks VR-specific measures.

Presentation Quality
•        Good organization with clear structure and flow,
•        Well-written technical content is easy to follow,
•        Figures clearly show system architecture,
•        Clear presentation of table results.

Impact and Innovation
•        High impact, especially novelty for LLM integration, low for VR application,
•        Additional impact for telepresence, limited for VR/AR field,
•        Practical applicability is demonstrated but needs real-world validation.

Concerns
VR Implementation
•        No details about the VR hardware setup
•        Missing information about:
        Headset specifications,
        Tracking system,
        Environmental design choices,
        User perspective configuration,
        Rendering approach.

Methodology
•        Participant VR experience criteria unclear,
•        Training protocol unspecified,
•        No VR sickness or comfort measures,
•        Limited exploration of immersion effects,
•        Missing VR-specific performance metrics.

Technical Integration
•        LLM integration well-documented,
•        VR platform choice not justified,
•        Limited discussion of:
        System latency,
        Computational requirements,
        Safety considerations.

Comments
The paper presents technically sound work but has several areas needing attention for acceptance to the VAM-HRI Workshop:

Strengths:
- Strong LLM integration framework,
- Clear statistical analysis,
- Novel approach to shared control,
- Good experimental design.

Primary Concerns:
1. VR Integration
   - Limited justification for VR use
   - Missing technical specifications
   - No discussion of VR-specific benefits
   - Unclear impact on user experience

2. Technical Implementation
   - Hardware specifications needed
   - Performance metrics incomplete
   - Environmental complexity is unclear

3. Experimental Design
   - Participant selection criteria
   - VR training procedures
   - System limitations
   - Failure case handling

While the paper shows innovation in LLM-based control, its treatment of VR aspects is superficial. The work appears more suited to a robotics venue focusing on shared control rather than a VAM-HRI workshop. However, the technical merit and potential impact on telepresence systems warrant consideration for acceptance.

The authors could significantly strengthen the paper by:
- Clarifying VR’s role in the research,
- Providing detailed VR implementation specifications,
- Analysing VR-specific interaction challenges,
- Exploring immersion effects on performance,
- Justifying VR as the chosen interaction medium.

---
Review completed: January 28, 2025

---

### Decision · Program_Chairs · 2025-02-26

Accept